# Welfare Assessment for Captive Anseriformes: A Guide for Practitioners and Animal Keepers

**DOI:** 10.3390/ani10071132

**Published:** 2020-07-03

**Authors:** Paul Rose, Michelle O’Brien

**Affiliations:** 1Centre for Research in Animal Behaviour, Psychology, Washington Singer, University of Exeter, Perry Road, Exeter EX4 4QG, UK; 2Wildfowl & Wetlands Trust, Slimbridge Wetland Centre, Slimbridge, Gloucestershire GL2 7BT, UK; michelle.obrien@wwt.org.uk

**Keywords:** waterfowl, wildfowl, welfare assessment, husbandry evidence, zoo, aviculture, animal behaviour, animal welfare

## Abstract

**Simple Summary:**

Zoological collections are constantly working to find new ways of improving their standards of care for the animals they keep. Many species kept in zoos are also found in private animal collections. The knowledge gained from studying these zoo-housed populations can also benefit private animal keepers and their animals. Waterfowl (ducks, geese, and swans) are examples of species commonly kept in zoos, and in private establishments, that have received little attention when it comes to understanding their core needs in captivity. Measuring welfare (how the animal is coping with the environment that it is in) is one way of understanding whether a bird’s needs are being met by the care provided to it. This paper provides a method for measuring the welfare of waterfowl that can be applied to zoo-housed birds as well as to those in private facilities. The output of such a welfare measurement method can be used to show where animal care is good (and should be continued at a high standard) and to identify areas where animal care is not meeting the bird’s needs and hence should be altered or enhanced to be more suitable for the birds being kept.

**Abstract:**

Welfare assessment is a tool to both identify welfare challenges and to evidence where current husbandry practices support positive welfare outcomes. Such tools are becoming more available and can be amended based on the nature of the facility and needs of taxonomic groups. Currently, welfare assessment has a strong mammalian theme, and some behavioural measures of welfare commonly applied to mammals do not translate well for other taxa. This paper provides a method for welfare assessment of Anseriformes; widely housed, diverse bird species kept under a range of management styles. A mixture of resource-based (i.e., determination of aspects of the physical environment or the bird’s physical appearance or activity) and animal-based (i.e., observations that equate to a bird’s feelings or personality characteristics) measures are integrated to enable a full review of potential predictors of welfare. The method provides a rapid and valid way for all personnel to collect information that evaluates quality-of-life experiences of the Anseriformes under their care. Explanations of key terminology are provided to enable repeatable and reliable assessment for all persons using the tool. Suggestions for follow-up actions are provided to emphasise why the welfare assessment process needs to be one of continual re-evaluation of animal care.

## 1. Introduction

Objective and repeatable measurement of animal welfare in zoos and collections of captive wild animals, across the course of an individual’s life stages and development, is becoming more and more common. Welfare is defined as the state of the individual as it attempts to cope with its environment [1] and encompasses physical, emotional, and behavioural components [2,3,4]. Welfare is experienced on a continuum and can change with time [5] across the course of an individual’s life and with differing physiological (e.g., in and out of the breeding season) and developmental (e.g., at sexual maturity) states [6]. Individual animal temperament and personality also influence how welfare state is expressed under prevailing conditions [7]. Repeated and valid assessment of animal welfare is a requirement of several zoo association accreditation processes, e.g., the European Association of Zoos & Aquaria, EAZA [8]. A welfare toolkit is a way that all those involved with the management of an individual, or of a species, or of an enclosure housing a species in captivity can collect information in a repeatable and valid way on the welfare state experienced by an animal/those animals at a specific time. An excellent example of this is provided by the British & Irish Association of Zoos & Aquariums (BIAZA) Animal Welfare Toolkit [9]. As welfare in non-human animals cannot be measured directly, inference of welfare state must be based on proxy measures that are biologically and ecologically sound to the species being observed.

Welfare assessment is focussed on resource-based and on animal-based approaches [10,11]. Resource-based measures can be scored accurately [12] as they review the provision that the animal is receiving [13]; these are often termed input measures. Animal-based measures (i.e., those giving an idea of output: how the animal is coping with the input) can be harder to assess [14] but provide much needed information on the suitability of the environment created [15,16]. Therefore, it is important to develop relevant qualitative, animal-based measures to further understand the emotional components of welfare in a particular species [17,18]. To be useful and practicable, a welfare assessment tool needs to be quick and easy for busy animal keepers to fill in accurately [9]. As zoo husbandry becomes more species-specific [19] and best practice guidelines are produced for individual taxa [20], so welfare assessment methods need to take into consideration the characteristics of individual species, how they may interact with their environment/captive husbandry regimes, and how they may demonstrate key aspects of wellbeing (e.g., behavioural and physical signs of coping or not coping). Many species of captive animals currently lack husbandry evidence for best practice care [21], with identification of positive welfare provisions and outputs part of this evidence need. Consequently, to assist with the development of regional and institutional collection planning and species-specific husbandry guidelines for a commonly housed group of birds, the Anseriformes, this paper details a method for welfare assessment whose results can be used to both evaluate the care birds receive and their responses to this care.

## 2. Aims of This Paper

The order Anseriformes incorporates around 180 species of wildfowl or waterfowl (ducks, geese, and swans) and the screamers (three species in the suborder Anhimae). The total global population of Anseriformes in the Species360 (Minneapolis, MN, USA) Zoological Information Management System (ZIMS) of registered zoos, as of March 2020 is 43,907 (a mixture of individual and group records) with representations from all main “types” of duck, geese, and swans (i.e., diving ducks, sea ducks, dabbling ducks, grey geese, and black geese) present [22]. Many populations are lacking in best practice guidelines and, as such, there is a need to objectively evaluate current husbandry standards to provide a basis for good care and good welfare for captive individuals. Given that Anseriformes can be kept in large flocks in relatively naturalistic enclosures, there is a need for a feasible, rapid, and repeatable welfare assessment method to be available for use with these species. Combined with birds housed in private aviculture, the keeping of captive Anseriformes is a large part of the exotic animal management industry. The lack of empirical, scientific evidence available for various impactful aspects of Anseriformes management (e.g., the effects of mixed-species housing and impacts of flight restraint) means a way of objectively and non-invasively determining how a bird is reacting to the care that it is provided with is required, which would be useful to private keepers and those working in zoological establishments.

Currently, there are no specific welfare assessment tools available for Anseriformes keepers to use to judge the quality of life of their animals under the conditions they are, at that moment in time, living in. Welfare assessment tools for domestic poultry *(Gallus gallus domesticus)* are available [23], but it may be tricky to translate such production-focussed work on domesticated birds to the wide array of wild species housed in exotic animal collections. A method to review the welfare of zoo animals that includes several Anseriformes, using a computerised assessment grid to score the physical, psychological, environmental, and procedural influences is available in the literature [24]. This assessment grid shows the importance of re-evaluation and re-examination of the animal’s environment and demeanour over time to ensure that welfare outcomes remain relevant to the individual and its wider social group. Welfare is a key consideration of conservation programmes (i.e., helping ensure a successful outcome) and with both EAZA and Association of Zoos & Aquariums (AZA) Anseriformes Taxon Advisory Groups (TAGs) working on Regional Collection Plans that can align, where appropriate, to the Conservation Planning Specialist Group’s (of the IUCN) One Plan Approach to Conservation [25,26]; a method for determining the responses of Anseriformes to their care and management is needed.

This paper presents examples of ecological evidence, taken from the published literature, to show where the basis for Anseriformes husbandry should originate from. It also provides an evaluation of species-specific natural history to help guide management challenges (e.g., enclosure style and flight restraint practices). Finally, a tool for Anseriformes welfare assessment is presented that enables evaluation of animal-based, resource-based, and behavioural measures of welfare for ducks, geese, swans, and screamers. Box 1 provides explanations of key terminology used within this paper to ensure the scientific basis of this welfare review is as accessible as possible.

Box 1Key terminology that explains the context of this paper.**One Plan Approach**: Integrated species conservation
planning, which considers all populations of the species, inside and outside their natural range, under all conditions of management, engaging all responsible parties and all available resources from the very start of any species conservation planning initiative. See: WAZA Magazine vol. 14 
https://www.waza.org/wp-content/uploads/2019/02/waza_mag_14.pdf
**Regional Collection Plan**: Describes which species are recommended to be kept, why, and how these species should be managed. Regional Collection Plans also identify which species need to be managed in European Endangered Species Programmes (EEPs). See: https://www.eaza.net/conservation/programmes/**Best practice guidelines:** Synthesise expert husbandry knowledge to make it widely available within and outside of the EAZA community. The guidelines show best practice standards, which EAZA zoos aim to achieve. See: https://www.eaza.net/conservation/programmes/**EAZA**: European Association of Zoos & AquariaSee: https://www.eaza.net/**AZA**: Association of Zoos & Aquariums See: https://www.aza.org/**Welfare**: The state of an individual as it attempts to cope with its environment, involving physiological, psychological, and behavioural measures. Adapted from Broom [1] and Green and Mellor [2].**Quality of Life**: The state of an individual’s life as perceived by it at any one point in time. It is experienced as a sense of well-being that involves the balance between negative and positive affective states. Quality of Life can be predicted by the fulfilment of basic and species-specific health, social, and environmental needs (and individual preferences for these) and is reflected in an individual’s health and behaviour. Adapted from Taylor and Mills [3].**(positive or negative) Affective state**: Concepts (e.g., feelings and emotions) that connect mental and physical processes. See: Mellor [4]**Behavioural expression**: Qualitative terms used to describe and summarise the different aspects of an animal’s dynamic style of interaction with the environment (e.g., bold, calm, nervous). See: Wemelsfelder, Hunter [7]**Fitness (biological)**: An organism’s biological fitness is dependent on its ability to survive and reproduce in a given environment.

## 3. Providing Evidence for Management Techniques: Adapting Universal Benchmarks for Specific Species

Factors that influence the attainment of positive states are the resources that are available around the animal for it to engage with (Figure 1). These resources and their availability and suitability (for the individual and the species) can be measured to provide the underlying explanation of the animal-based measures of welfare (e.g., behavioural expression, personality changes) that result from them.

Due to evolutionary and ecological differences between taxa, general welfare assessment models need to be adapted for use at a specific taxonomic level. The specific evolutionary pathway that Anseriformes have travelled means the performance of some behaviour leads to fitness benefits to the individual and to the population (in the long term). For example, domestic duck *(Anas platyrhynchos domesticus)* welfare considerations require birds to have access to water troughs that enable complete immersion of the bird’s head [28]—thus linking together an evolutionary requirement with a health and welfare output. Waterfowl housed without adequate bathing facilities develop poor plumage conditions (e.g., “wet feather”) that can lead to chronic health compromises [29,30].

With captive wild birds in the zoo, such baseline measures need to be extended to not only provide sufficient opportunities for basic maintenance behaviours (i.e., preening and bathing) but enable water access to provide for a range of fitness-promoting behaviours across the course of the individual’s life in the zoo. As these birds will grow, develop, mature, and enter senescence in a human-controlled environment, and live for much longer than the individuals for which agricultural guidelines are provided, welfare must be considered across all life stages. The ecological need of eiders (*Somateria* sp.) or pochards (*Aythya* sp.) to dive, or shovelers (*Spatula* sp.) to filter-feed, or swans (*Cygnus* sp.) to up-end whilst foraging [31,32,33] must be considered in the area and depth of swimming water provided if such behaviours are to be maintained across generations in captivity. We cannot always be certain that performance of all wild behaviour is essential for an excellent quality of life [34] and whilst the conservation of adaptive behavioural traits is essential to meeting the conservation, education, and research roles of that species in captivity, how the expression of such behavioural traits underpins welfare inferences and the assumptions we need to make on the importance of their performance remains tricky to define.

A key consideration of assessing bird welfare is to think about the differences in how birds can display natural behaviours that have a specific evolved function or fitness benefit. Flight restraint (e.g., non-reversible pinioning or temporary feather trimming) is still commonly practised to enable the keeping of Anseriformes in open-topped exhibits. The arguments for flight restraint centre around the provision of wide, open spaces in which birds can behave naturally and that as Anseriformes moult their flight feathers completely each year and have other mechanisms for movement as well as flight (swimming, diving), they are less impacted by flightlessness compared to other members of the class Aves. The contradictory arguments centre on the ethical debate of removal of a natural, innate activity, the lack of behavioural choice available to the bird, and the irreversible nature of some flight restraint methods such as pinioning. The ethical debate around flight restraint is complex and emotive, but from a welfare science perspective, there is little to no information available on the welfare impacts of flight restraint on Anseriformes behaviour or how flight restraint interacts with the welfare impacts of enclosure styles, areas, and sizes, for example.

We can attempt to infer, from an aspect of a bird’s ecology and behaviour, how welfare may be impacted if an activity cannot be performed in the same way in captivity, if the activity in question is important to the species’ way of living in the wild. Hence, we may be able to make assumptions on which forms of husbandry may be more appropriate for some species rather than others based on this inference of ecological difference. For example, taking an aspect of wild Anseriformes behaviour that differs markedly between species, i.e., time spent perching off the ground, as a potential welfare measure, we could provide some insight into which species of Anseriformes are more suited to being managed using flight restraint compared to those that may do better in aviary housing (Figure 2). Therefore, an evidence-basis for the most appropriate housing style and enclosure at a species-specific level can begin to be pieced together. Ideally research, a combination of in situ and ex situ focussed study, should be undertaken to verify this assumption to enable more certainty about if perching is a clear indicator of welfare state.

Table 1 provides an explanation for the species of Anseriformes chosen for inclusion in Figure 2, based on their ecology and habitat choices and evidence of degree of perching or arboreal behaviour in the wild. Taking the whistling ducks as an example, whilst the alternative common name for these birds is “tree ducks”, not all species perch or are often seen in trees or choose to nest in trees [43]. The white-faced, black-bellied *(D. autumnalis)* and West Indian *(D. arborea)* whistling ducks are highly arboreal whereas the wandering *(D. arcuata)* and fulvous *(D. bicolor)* whistling ducks are much more aquatic and terrestrial [32,48,49]. Therefore, we must consider the individual species of whistling duck and its behavioural needs to perch off of the ground when assessing captive welfare and not make a sweeping judgement that all species of whistling duck must be provided with perching off the ground. However, as all whistling duck species are not fully cold tolerant [49] and mortality in colder climates can be reduced by keeping birds fully winged when housed in outdoor accommodation [43], the local environment and location of the zoo and the aviary style of the bird needs to be considered alongside of taxonomy when fully assessing the suitability of husbandry. Here, whistling ducks sit in the middle of Figure 2 as an example of a species whose needs can be difficult to generalise in captivity as ecological and natural history information can be a challenge to decipher.

As populations of private Anseriformes are often integrated into those of captive collections, for example, for ensuring the health and diversity of bloodlines of small populations or for increasing the holding capacity for particular species, individuals in private care can also be welfare assessed to help improve housing and inform and educate owners. The relevance of all bloodlines of captive Anseriformes to long-term conservation initiatives (e.g., wider integration of ex situ populations into One Plan conservation approaches) empowers all those keeping captive birds to welfare score their animals, ensuring that populations can thrive rather than just survive [21] and key adaptive traits are conserved across all populations of all individuals.

Evidence, albeit anecdotal, is available to enable Anseriformes keepers to make educated guesses about the most appropriate housing methods for waterfowl. For example, information on the free-to-access Wildpro website [50] states that “Perching ducks and ‘geese’ are generally happier maintained fully-flighted if possible, for example in an aviary for the smaller species, or under flight netting”. The ecology of the bird (i.e., the description of perching duck) is emphasised as the evidence for why an enclosure needs to provide height and opportunities for free flight. Likewise, an increased degree of cold sensitivity and likelihood of frostbite or cold-related mortalities (plus other associated infections) occurring in captivity appears to correlate with those species who prefer to perch off the ground [49,51,52,53].

## 4. A Tool for Anseriformes Welfare Assessment

Time minimal welfare assessment methods are essential for busy animal keepers to complete. The information obtained from welfare assessments needs to be fed back into an action plan that provides the route of change in animal management to rectify any specific welfare issues that the assessment has noted. The welfare assessment methods presented in Appendix A have been designed to assist keepers in rapid and easy to compile data collection on their birds. Appendix A methods were based on trials of these welfare scoring techniques on whistling swans, and spectacled eiders *(Somateria fischeri)* at different Wildfowl & Wetlands Trust (WWT) centres, as well as on mandarin *(Aix galericulata)* and North American wood duck *(A. sponsa)* in a private collection. Personality descriptors were devised and trialled by a waterfowl-specific veterinary surgeon and this paper’s author and trialled for repeatability on the species noted above. Behavioural measures were devised from the common state behaviours, i.e., long-duration components of a daily time budget [54], that Anseriformes are known to display based on published literature [55,56] or observation of birds by experienced keepers.

### 4.1. Example Results Output on Bird Behaviour and Personality

An example of the application of this welfare assessment method is provided below. This example output was conducted by the authors of this paper on the usefulness of this welfare scoring method for captive swans held at WWT Slimbridge Wetland Centre to outline different observer reliability testing and personality scoring for an individual bird (Table 2). Observations per swan took around five minutes to complete. Interobserver reliability (IOR) testing was conducted on the personality descriptors prior to the methods being trialled and again once the methods had been used for welfare scoring observations. Interobserver reliability was calculated as: number of agreements/(number of agreements + the number of disagreements) × 100 [54].

For the testing of the methods (“pre-obs”), on two birds of a closely related subspecies (Bewick’s swan, *Cygnus columbianus bewickii*) with the same behaviour pattern [57,58], both observers scored 76% reliability. For the first three observation points, IOR agreements were 100%, 71%, and 100%. Photos of what the bird was doing and how it presented to the observers for these three observations are provided to show the view that the two observers had (Figure 3).

The physical feature (Section 1) for the welfare assessment tool gauged from these observations is that the bird is on land and with conspecifics. The plumage and body condition of the bird is excellent (score a 5) and ease of movement on land cannot be judged (“not seen”) for Obs 1 and Obs 3 but would be excellent for Obs 2. Ease of movement in water would be “not seen”. This species of waterfowl does not go into an eclipse plumage [59] and the birds are not in moult during the season in which these observations were conducted. Behaviour of the swan was classified as Standing, Resting/sleeping (Obs 1); Vigilance, Standing, (Obs 2); and Resting/sleeping (Obs 3). When assessing the relaxed state of the group overall (Section 2, animal features), both observers agreed that the birds looked Comfortable (Obs 1 and 3) and Agitated or Unsettled (Obs 2). These descriptions of behaviour patterns would be compared to the environment around the bird as well as the bird’s condition/plumage quality/movement characteristics to make a judgement on its welfare state.

The output of a factor analysis (conducted in SPSS v. 26 [60]), Figure 4, shows that two observers trained in the same method of scoring for behavioural expression (for the personality part of Section 2, animal features) can produce consistent results, grouping descriptors of behavioural expression together in the same way. Record keeping of changes to behavioural expression over time is useful for the private Anseriformes keeper to allow further investigation of the reasons for any untoward alteration to what may be considered the “normal” personality for a bird. For a zoological collection employing a welfare officer, formal analysis of behavioural expression may be more readily undertaken to help document welfare state as bird populations or husbandry routines develop or change alongside of the zoo’s personnel and collection planning.

A notable point of consideration shown by Figure 4 is the descriptor “bored”, which has been classified similar to the “relaxed” and “restful” characteristics by one observer and with the “upset” and “annoyed” characteristics by the other. This shows the importance of revisiting welfare scoring methods regularly and checking that all observers understand what that measure of psychological state may be in that individual or species. These results were used to simplify the personality options in the final welfare assessment toolkit, removing some ambiguous descriptors and providing more detail in the explanation of what to score.

The factor analysis shows that the first three components account for 55%, 16.5%, and 6.7% of the variance explained and the rotated component matrix (Table 3) shows the factor loadings (factors less than 0.3 excluded) for these first three components to show which personality descriptors correlate within which principal component.

### 4.2. Practicalities of Using This Welfare Assessment Tool

The questions posed in Appendix A are designed to encompass evaluation of all types of environments that Anseriformes are housed in, as well as capture as many predictors of or influences on welfare state as possible. Questions can be used at an individual level and provide a way for keepers to keep track of changes to a bird’s quality of life over time. Outputs can be fed back into management plans for specific species within the collection and enable informed changes to housing and husbandry regimes to be implemented based on what the birds are “telling” their keepers. The template is flexible to all, allowing for questions to be edited or added based on the individual needs of the animal collection. For example, an institution that houses Anseriformes in mostly mixed species covered enclosures can tailor its welfare assessment more specifically by focussing on the influences of multi-species holdings rather than on pest species or flight restraint, for example. A suggestion for how to use the Anseriformes welfare assessment method to implement changes to management and how to continually evaluate bird management regimes is provided in Figure 5 with a practical, species-specific example outlined in Figure 6.

A standardised approach to welfare assessment means that evaluation of past welfare scores, alongside of current information from the animal, can be performed with confidence because methods will have been reliably applied by all persons completing an assessment. The cycle of observation, review, evaluation, and re-observation (Figure 5) is possible due to the collation of past records and the standard approach to scoring. The welfare assessment method has been designed based on industry practice that shows that a standardised scoring method is key to correct and reliable assessment of all parameters that are influencing attainment of positive welfare states [17,61].

Whilst the performance of natural behaviour is only one aspect of good welfare, it is helpful for all animal collections holding Anseriformes to provide key information on ecological needs (as per Table 1) to all personnel involved in animal management so that husbandry can continually be evaluated and reviewed alongside of bird behavioural needs and evolutionary history. Therefore, the welfare scoring performed by the animal collection will be the best possible fit to what the bird has evolved to do. This provides a sound starting point for the animal to thrive, rather than just survive, when housed in human care, and further enables refinement of all areas of husbandry to promote positive welfare states.

Information collected from a welfare assessment must be used in a continual cycle of evaluation and re-assessment of the management plans for that animal or population of animals (Figure 5). Identification of areas of husbandry or enclosure change that may be beneficial to bird welfare can be reviewed, and change implemented, based on this evidence that has been gathered. Using ecological information in a welfare assessment helps to judge the quality and suitability of the physical resources that will influence aspects of the bird’s welfare state as well as guide evaluation of any impacts of other animals around the bird to ensure that management practice continues to be fit for the purpose. Figure 6 provides an example of how judgements on bird welfare and evaluation of husbandry suitability are in part grounded in knowledge of a species’ ecological needs, to provide an evidence-based approach to husbandry change where needed to promote positive welfare states. The captive environment that does not pose all of the same challenges as does the wild, and, therefore, not all facets of a species’ behavioural ecology may be appropriate or relevant to assessment of bird welfare. A lack of performance of some forms of natural behaviour, therefore, is not automatically a sign of impoverished welfare in the captive environment.

The questions posed for welfare assessment in Appendix A can be altered according to species, population, zoo, or region, for example, by exchanging in and out resources where appropriate to that species’ needs and editing the identification of wild/feral animals that are likely to be in or around a specific waterfowl enclosure. Consideration of bird time-activity budgets and the temporal effects on activity also need to be considered. Waterfowl are generally more active in the morning and later afternoon [62,63,64], with some species being cathemeral [65] and others having specific nocturnal activity patterns [66,67]. Therefore, judgement of behavioural expression (i.e., when individuals may appear to look lethargic or restless or disinterested) needs to made with knowledge of what that species is likely to be doing at the time the welfare assessment is taking place.

### 4.3. Guidance for Using the Welfare Assessment Tool

As previously mentioned, the tool is flexible and allows for individual adaptation to the needs of the institution, collection, or population of birds. To help with completion of the tool, guidance on each section is provided in this section. For parts of the welfare assessment tool where the answer might be “not seen”, it is important to recheck and re-assess as soon as possible to ensure that all variables with the potential to influence an individual’s welfare state have been measured as accurately as possible. Where behaviour and activity are scored, trials of the method have shown that it works well as an instantaneous tool so long as it is applied consistently (i.e., not just once per year), and an accurate picture of an animal’s behaviour can be drawn from a short period of observation per enclosure.

For Section 1 of the tool (Physical features), recording of where the bird is in the enclosure (i.e., on land or water) and what the bird’s social state is helps to evaluate the normality of other behaviours that are assessed in this section. Ease of movement and plumage condition can provide a clear illustration of current underlying health and physiological state and be linked back to behaviour and to time of year; and if scoring is used regularly, changes in locomotion and in physical condition can be tracked and improvements or deterioration with environmental and husbandry alterations can be documented and any need for intervention evidenced. Basic categorisation and evaluation of behaviour patterns at the time of assessment again provides information on underlying motivational and physiological states and changes to what may be considered “normal” behaviour for an individual or population at a specific time of day or season can then be recorded and further investigated. The top sections of scoring visitor and keeper presence can be useful for those completing welfare assessments in a zoological collection where the proximity of members of the public and the zoo’s workforce may influence where birds are likely to be seen in the enclosure and affect what the animals may be doing.

For zoos attempting to quantify the emotional state of their birds, for example, to comply with accreditation requirements or to evidence positive states in animals kept under a new husbandry scheme, a basic way of determining “mood” from behavioural expression (see Box 1 for definition) is provided in Section 2 of the tool (Animal features). These descriptors, which help illustrate mood or feeling, can be scored by different observers so long as the same definition of the behavioural expression is adhered to by all persons involved in welfare scoring. For each of the descriptors, a definition and description of what to look for when performing the welfare assessment tool is provided in the explanatory notes to help with consistent welfare scoring. Results from Section 2 can be used to do a formal qualitative behavioural assessment, QBA (if so required by the zoo) and excellent examples of this method are provided in the literature [17,68,69,70]. A summary of how QBA is useful to captive animals, with wildfowl examples, is provided in [71].

For those wishing to simply document the demeanour of their birds and how this changes with time, scoring of behavioural expression can be used to support changes in the bird’s physical appearance and behaviour and evidence when husbandry interventions may need to be improved to ease management (e.g., forewarning of when individuals are likely to become aggressive with season and when more resources should be provided to mitigate this). Individual bird behavioural expression scores can be kept in home records should they be needed to decipher activity or changes to demeanour over the course of a bird’s lifetime.

The final part of the tool, Section 3 (Enclosure features) provides a way of assessing how the resources provided to the bird, as well as how the bird can access and interact with said resources, influence welfare and hence the characteristics noted in Section 1 and Section 2. Particularly useful for larger collections with naturalistic exhibits, identification of native or feral species that enter the bird’s enclosure and utilise the space will help determine how changes to husbandry and how pest scaring or determent schemes (e.g., such as changing feeding schedules or altering barriers into the exhibit) positively impact on the activity and on the behavioural expression of the collection animals. Likewise, if poor scores for natural behaviours from Section 1 and consistent scoring from Section 2 are suggestive of lethargic or apathic states, then the overall husbandry regime may not be ecologically relevant to the species being housed and changes to husbandry, based on lower scores or identification of absent resources (as identified in Section 3), can be implemented. Follow-up welfare assessment should then be performed to see the effect of husbandry change on bird quality of life.

It is essential to keep in mind the effects of daily husbandry (e.g., feeding times) on bird activity and complete observations during daily maintenance times as well as at times when husbandry and management tasks are not being completed. Consequently, the recording of date and time, to link scoring back to external influences on bird welfare, is important to the accurate evaluation of scores.

### 4.4. Parameters of Importance to Individual Species

Whilst this tool has been specifically designed to be flexible across different facilities and for different species, the specific requirements of some species need to be considered as essential or obligate to the attainment of positive welfare when completing an assessment. Such essential welfare requirements would be based on the key aspects of natural history of ecology that promote the performance of behaviour patterns linked to appetitive behaviour patterns. Key examples (Table 4), in a non-exhaustive list, of applying the welfare toolkit to specific Anseriformes groups and identifying non-negotiable aspects of husbandry to promote good welfare are provided below.

### 4.5. Combining Measures from Across the Welfare Assessment Tool

Combining the measures from each part of the tool is important to provide an overall picture of the bird’s welfare state. Birds can provide a challenge when it comes to inferring health and wellbeing from observation alone, as many symptoms of poor health could be masked by plumage or hidden as a survival mechanism (i.e., to reduce attention from predators), and repeated measurement is needed to provide an accurate picture of individual animal wellbeing [24]. Keepers need to be aware of the individual and species-specific needs of the Anseriformes in their care to make informed decisions on what the welfare scores are suggestive of. Enclosure features may appear suitable, but birds are unable to fully exploit them. Birds may appear in good physical condition because diet is suitable but lack stimulation from the enclosure and thus display limited or frustrated time budgets.

Birds may appear to be in excellent physical and behavioural conditions in a poor enclosure as they are currently coping with the situation they are in, but further physiological or psychological change (e.g., requirements above normal maintenance) may cause extra stress that results in determinantal changes to plumage condition or body condition or the increased performance of negative behaviours (e.g., pacing or unwanted aggression). An individual bird that displays personality characteristics suggestive of poorer welfare state (e.g., bored or depressed) may still be in good overall condition if it has access to a suitable diet and clean water for bathing. Therefore, consideration of environmental or social changes to improve behavioural expression would be needed without altering key aspects of daily husbandry that are providing for some positive outcomes. The continued review of the enclosure and its features, alongside of the bird’s individual characteristics, will provide the most accurate illustration of how environmental and animal variables interact to influence an individual’s welfare state.

## 5. Conclusions

This paper is designed to provide a guide to help assess the quality of life experienced by a group of birds commonly housed in public and private animal collections. Output of the welfare assessment method should be collated and stored, and shared with all those responsible for bird care, to enable evaluation of husbandry across time. Some areas of Anseriformes husbandry is lacking a strong evidence basis and the more that individual bird welfare is examined and measured in a repeatable fashion, the more that good aspects of husbandry practice can be disseminated across all institutions holding these birds. The importance of many captive populations of Anseriformes to species conservation outcomes supports the need for a species-specific method of welfare assessment to be freely available for use by all keepers of these birds.

## Figures and Tables

**Figure 1 animals-10-01132-f001:**
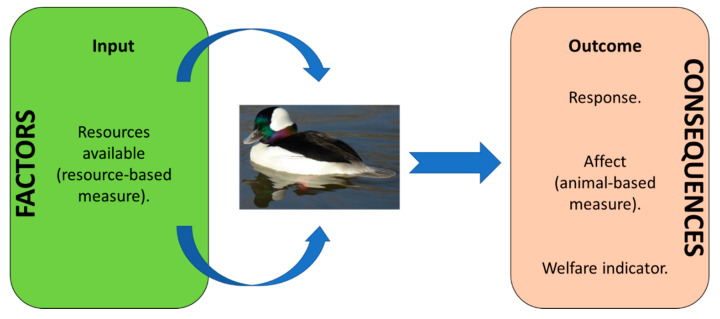
Integration of resource- and animal-based measures for captive waterfowl, adapted from the European Food Safety Authority (EFSA) Panel on Animal Health & Welfare [27]. Resources are those provided via management and those found within the environment.

**Figure 2 animals-10-01132-f002:**
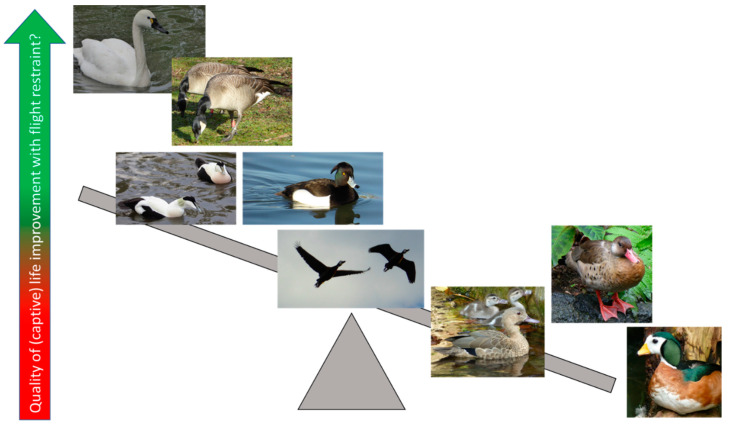
Anseriformes species that are often seen to perch (right of the diagram) compared to those that do not perch (left of the diagram). Measurement of time spent perching or sitting off the ground could guide management practices around flight restraint. The whistling swan (top left) has a wild ecology that suits it more to containment by flight restraint compared to the African pygmy goose (bottom right), whose evolutionary adaptations enable it to perch and nest off the ground, based on these bird’s ecological differences as outlined in Table 1.

**Figure 3 animals-10-01132-f003:**
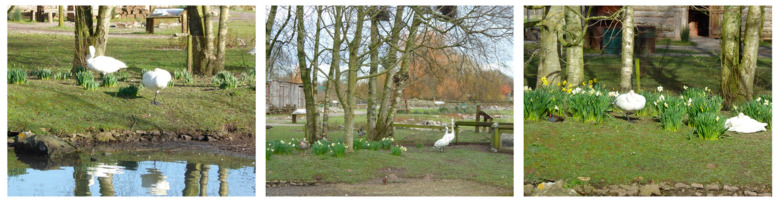
Photos of the whistling swans from the three observations (left to right: Obs 1–3) in Table 2 to illustrate the view of the birds during welfare assessment. The male bird that this example assessment was completed for is located at the front (Obs 1), rear (Obs 2), and right (Obs 3) of the photographs.

**Figure 4 animals-10-01132-f004:**
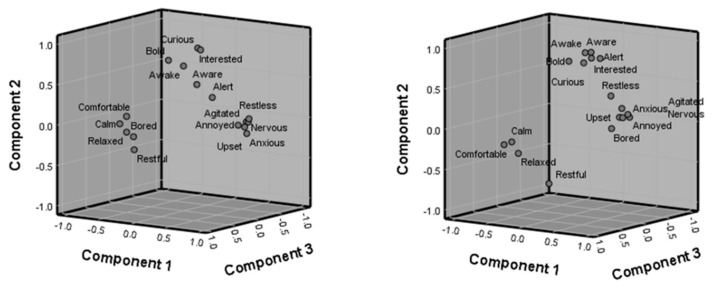
Component plots from a factor analysis using data from two different observers scoring the behavioural expression (psychological welfare) of a male whistling swan over a one-year period. Groupings of descriptors that explain the same amount of variation within the observations appear together.

**Figure 5 animals-10-01132-f005:**
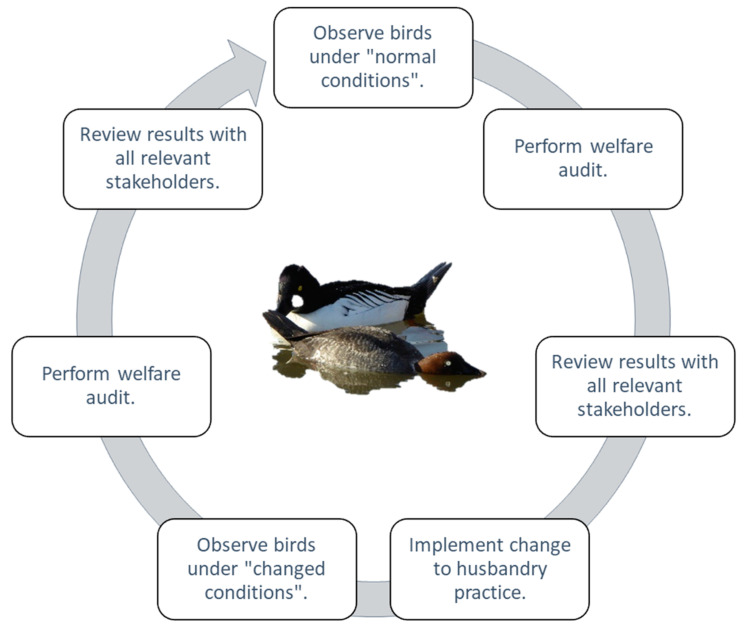
A suggested plan for use of this welfare assessment method for captive birds to provide cyclic evaluation and re-assessment of Anseriformes husbandry. “Normal conditions” means the husbandry experience of the bird before a change to management, social group, or environment has been implemented. “Changed conditions” means observation of the bird after an alteration to husbandry, housing, or management after review of a welfare audit. “All stakeholders” denotes those personnel who have a duty of care to the individual duck in that animal collection.

**Figure 6 animals-10-01132-f006:**
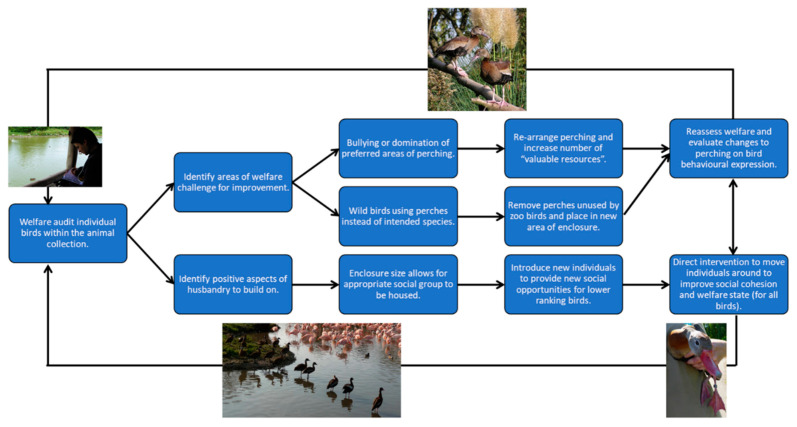
Conducting a welfare assessment on an example species of waterfowl, in this case a whistling duck, using knowledge of ecological needs (i.e., perching and flock-living) to guide evaluation of biologically suitable husbandry and to identify areas of husbandry improvement and welfare challenge. Photo credit (perching whistling ducks): J. Harteman (used with permission).

**Table 1 animals-10-01132-t001:** Examples of ecological information used to understand a species’ needs for being off the ground in captivity, focusing on the Anseriformes depicted in Figure 2.

Species	Ecology
Tundra (whistling) swan, *Cygnus columbianus columbianus*	Open coastal wetlands, shallow pools, lakes, and rivers. Marshland and coastal grasslands in winter. Nests on the ground; islands in tundra pools a preferred setting. Feeds by walking/grazing as well as by up-ending [32] and does not travel far to locate food around breeding areas [35].
Cackling goose, *Branta hutchinsii*	Open tundra grasslands and nests on the ground, usually near water [36]. Feeds by walking/grazing but will also forage aquatically [37].
Common eider, *Somateria mollissima*	Highly maritime, occupying marine environments all year round and specialised for diving [38]. Weight and wing loading can reduce the flying abilities of this duck, grounding it in calm weather when no wind is available to aid take-off and when females are carrying extra weight of eggs [38]. Eiders are principally aquatic, with terrestrial locomotion limited by diving adaptations [39].
Tufted duck, *Aythya fuligula*	Coastal lagoons, shorelines, gravel pits, and sheltered ponds are favoured feeding sites, where this duck feeds predominantly by diving for molluscs and crustaceans. Nests are sited close to or on highly vegetated marshes and lakes [40,41].
White-faced whistling duck, *Dendrocygna viduata*	Subject to unpredictable local movements and to being highly social [32]. These ducks can make flights between feeding and loafing sites, spending the majority of the day loafing, preening, or sleeping [42]. Nesting in trees and on the ground is noted [32,43].
Madagascar teal, *Anas bernieri*	Known to perch and nest in hollow trees [44]. Found in a range of wooded to heavily vegetated wetlands as well as more open waterways [45].
Brazilian teal, *Amazonetta brasiliensis*	Prefers a habitat of pools, streams, marshes, and small lakes in dense woodland where it commonly perches on branches overhanging the water [31,46]. Wide variety of nesting sites, both on water as well as in trees and on cliff edges [31,32].
African pygmy goose, *Nettopus auratus*	Short legs mean this bird is ungainly on land [40]. Nests off the ground, 10 to 20 m high [32,47]. Perches on branches over-hanging water [40].

**Table 2 animals-10-01132-t002:** Example of personality scoring for an individual swan. The bird’s personality was scored from 0 (not describing the bird’s behavioural expression at all) to 100 (completely describing the bird’s behavioural expression). Cells in light grey indicate those where observers did not agree.

Session/Observer		Agitated	Alert	Annoyed	Anxious	Awake	Aware	Bold	Bored	Calm
Pre-obs	A	90	100	70	90	100	100	70	0	20
	B	85	100	60	90	100	100	70	0	0
Obs 1	A	0	0	0	0	0	0	0	0	100
	B	0	0	0	0	0	0	0	0	100
Obs 2	A	70	100	10	80	100	100	90	0	30
	B	70	100	0	70	100	100	90	0	20
Obs 3	A	0	0	0	0	0	0	0	0	100
	B	0	0	0	0	0	0	0	0	100
		Comfortable	Curious	Interested	Nervous	Relaxed	Restful	Restless	Upset	
Pre-obs	A	20	60	70	90	0	0	100	80	
	B	0	60	70	90	0	0	100	80	
Obs 1	A	100	0	0	0	100	100	0	0	
	B	100	0	0	0	100	100	0	0	
Obs 2	A	40	100	100	50	10	0	80	50	
	B	30	100	100	50	0	0	80	50	
Obs 3	A	100	0	0	0	100	100	0	0	
	B	100	0	0	0	100	100	0	0	

**Table 3 animals-10-01132-t003:** Rotated components matrix showing the factor loadings for each descriptor.

Descriptor	1	2	3
Anxious	0.918		
Nervous	0.911		
Calm	−0.893		
Comfortable	−0.866		
Relaxed	−0.857	−0.319	
Upset	0.856		
Agitated	0.834		
Restless	0.811		
Annoyed	0.769		
Restful	−0.634	−0.578	
Interested		0.883	
Curious		0.855	
Awake		0.833	
Bold		0.789	
Aware	0.359	0.770	
Alert	0.554	0.674	
Bored			0.872

**Table 4 animals-10-01132-t004:** Examples of essential enclosures or husbandry requirements for select Anseriformes groups to guide the completion of a welfare assessment.

Species Group	Key Welfare Need	Reference
Screamers	Some degree of off-ground perching. Correct social group (pair bond). Straw/shelter provided in winter to protect extremities from the cold.	[72,73,74]
Swans	Opportunities for natural foraging (i.e., feeding at depth and grazing). Opportunities for pair bond formation. Ease of access in and out of water especially important for black-necked swan (*Cygnus melanocoryphus*).	[73,75,76]
True geese	Opportunities for natural foraging (i.e., grazing). Correct social group to allow for social interactions.	[73,75,77,78]
Shelducks/sheldgeese	Appropriate social group (reduce aggression to other birds, ensure territory establishment). Opportunities for natural foraging (i.e., grazing).	[73,75]
Dabbling ducks	Appropriate sex ratio (i.e., to ensure reduced or removed female harassment). Opportunities for natural foraging (i.e., grazing for wigeon species, filtering in open water for shovelers).	[29,30,75]
Diving ducks	Free-flowing, non-stagnant water. Deep water for diving.	[75]
Tree ducks	Some degree of off-ground perching. Correct social grouping (e.g., pair bond formation for whistling ducks). Straw/shelter provided in winter for tropical species.	[49,51,75,79]
Sea ducks including steamer ducks	Free-flowing, non-stagnant water. Deep water for diving. Single species housing essential for steamer ducks.	[73,75,80,81,82]
Stiff tails	Ease of access to and from swimming water. Deep water for diving.	[75,83]

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
