# Peer review of "Welfare Assessment for Captive Anseriformes: A Guide for Practitioners and Animal Keepers"

_animals, 2020, doi:10.3390/ani10071132_

Round 1

Reviewer 1 Report

Welfare assessment for captive Anseriformes: A guide for practitioners and animal keepers

Welfare assessment tools are becoming increasingly common for a range of animals in a wide range of habitats, including the wild, and this paper provides one to assess Anseriformes welfare. I have really enjoyed this paper, and hope my comments will help the authors improve the manuscript even further. I have two major comments to make:

First, the authors have tried to create a tool that can be used on a wide range of species, and although this has its advantages, it also has its disadvantages. For me, the main disadvantage is that the tool provides no indication of how important each parameter is as an indicator of welfare. Then authors have not addressed this because, quite rightly, this is different for different species. However, the concern is that keepers may then simply focus on the parameters that they think are important, and which are “good” in their system, and consider the ones that are “bad” in their system as less important. (I don’t wish to criticise keepers with a broad brush, but I’ve come across some that focus on the good aspects of their systems and dismiss indicators of poor welfare as either unimportant or consider it a problem with the animal itself). This defeats the object of the tool which is to try and provide a more “objective assessment” of welfare. I think some guidelines, perhaps listing which parameters are important (or perhaps “non-negotiable”) for which species, may be helpful.

Second, I think the paper requires some instructions on how the measures obtained for each parameter of the tool should be combined to provide an overall assessment of welfare. There is no information on how the measures for each parameter should be combined. For example, if the “animal features” scores are fantastic, but the “enclosure features” are terrible, then what is the implications for welfare? Similarly what would great physical features but terrible animal features tell us? I would suggest a section before the conclusion is necessary to provide some instruction on this matter.

Specific comments:

Lines 75-90. In the interest of brevity, I don’t think this section is necessary. Welfare assessment tools are becoming more common, and you’ve made your point in the previous 3 paragraphs as to their benefit.

Figure 1 in line 107. Should be “Affect”, not “Effect”?

Section 2 and 2.1. I really enjoyed section 2.1, but I didn’t think it fitted under the “Aims of this paper” heading. It gives the impression that the aim is simply to quantify the welfare impact of preventing flight. I think the simplest solution may simply be to make 2.1 its own heading, i.e. 3.

Line 228-240. I welcome the authors’ attempt to validate the welfare assessment tool. However, exactly what was done, and all the results need to be presented. This is critical to inform the reader of the validity of your tool- it is not sufficient to simply provide the component plots for one species. In this section I would have expected to see far more analysis such as, intra and inter-observer reliability scores, and possibly PCA or other such analysis.

Appendix 1: Item 3. Typo, “in place THAT provides…”.

Lines 254-259, Lines 270-277. In section 2.1, the authors discuss that natural behaviour offers a good starting point to consider welfare, but that in captivity, some birds may not need to display natural behaviour in a captive environment where they don’t encounter the same challenges as in the wild. So I was a bit disappointed that lines 254-259 and 270-277 give the impression that improving welfare is simply about letting the birds do their natural behaviour. I don’t think this was your intention, but it comes across like that when I read it. Perhaps you could re-consider writing these sections to make clear that simply allowing animals to do natural behaviour may not be necessary to ensure good welfare.

Author Response

Reviewer 1

Welfare assessment tools are becoming increasingly common for a range of animals in a wide range of habitats, including the wild, and this paper provides one to assess Anseriformes welfare. I have really enjoyed this paper, and hope my comments will help the authors improve the manuscript even further. I have two major comments to make:

First, the authors have tried to create a tool that can be used on a wide range of species, and although this has its advantages, it also has its disadvantages. For me, the main disadvantage is that the tool provides no indication of how important each parameter is as an indicator of welfare. Then authors have not addressed this because, quite rightly, this is different for different species. However, the concern is that keepers may then simply focus on the parameters that they think are important, and which are “good” in their system, and consider the ones that are “bad” in their system as less important. (I don’t wish to criticise keepers with a broad brush, but I’ve come across some that focus on the good aspects of their systems and dismiss indicators of poor welfare as either unimportant or consider it a problem with the animal itself). This defeats the object of the tool which is to try and provide a more “objective assessment” of welfare. I think some guidelines, perhaps listing which parameters are important (or perhaps “non-negotiable”) for which species, may be helpful.

Thank you for the comment. We appreciate what you have mentioned here but there are currently no guidelines provided for birds of any form in captivity to assess and score welfare. We have cited the one paper that uses an animal welfare assessment grid to look at some measures of bird welfare in the zoo, but these are not evaluated specifically by species or resource. We have provided information on how the different aspects of the tool informs welfare and we have stated that these parameters help underpin good welfare in a species specific manner, for example by giving the option to not record if not relevant for that species.

Wildfowl (ducks, geese, swans) and screamers have key biological requirements that must be met in captivity. We have attempted to include information on what is important for key species within the document to help clarify that this is over-arching for all Anseriformes, but some aspects of care are specifically important for some species of this taxonomic group. This is included as a new section, 4.4. Parameters of importance to individual species where there is a table of pointers to guide the completion of the toolkit.

We hope that these changes and the inclusion of new information provide a suitable response to the useful points that have been raised here.

Second, I think the paper requires some instructions on how the measures obtained for each parameter of the tool should be combined to provide an overall assessment of welfare. There is no information on how the measures for each parameter should be combined. For example, if the “animal features” scores are fantastic, but the “enclosure features” are terrible, then what is the implications for welfare? Similarly what would great physical features but terrible animal features tell us? I would suggest a section before the conclusion is necessary to provide some instruction on this matter.

Thank you for the comment. This is a useful suggestion and we have included more information on this to guide with the usefulness of the tool for those wishing to use it. We have written a new section, 4.4. Combining measures from across the welfare assessment tool, before the conclusion. We have tried to include more information across the paper on how the results can be used, without trying to be prescriptive.

Specific comments:

Lines 75-90. In the interest of brevity, I don’t think this section is necessary. Welfare assessment tools are becoming more common, and you’ve made your point in the previous 3 paragraphs as to their benefit.

We have removed this section as per the suggested comment. However, based on the comments from another reviewer, we have moved one small part (about collection planning) to another part of the paper to enhance the clarity of another section.

Figure 1 in line 107. Should be “Affect”, not “Effect”?

We have edited this figure accordingly.

Section 2 and 2.1. I really enjoyed section 2.1, but I didn’t think it fitted under the “Aims of this paper” heading. It gives the impression that the aim is simply to quantify the welfare impact of preventing flight. I think the simplest solution may simply be to make 2.1 its own heading, i.e. 3.

Edited according the suggestion above.

Line 228-240. I welcome the authors’ attempt to validate the welfare assessment tool. However, exactly what was done, and all the results need to be presented. This is critical to inform the reader of the validity of your tool- it is not sufficient to simply provide the component plots for one species. In this section I would have expected to see far more analysis such as, intra and inter-observer reliability scores, and possibly PCA or other such analysis.

We have included more information here and expanded on the output of the factor analysis. We have provided information on interobserver reliability and we have explained how we score personality descriptors and animal behaviour. We have included more data on the personality scoring (including an explanation of how to score) and we present some key identifiers of physical condition too. We are trying to present a practical tool that can be applied to a range of husbandry and management situations where these birds are kept, so we have deliberately not attempted to provide a scientific analysis of welfare for fear of people not attempting to do welfare scoring in case they feel it might be too complicated. The idea for this tool is based on the BIAZA Animal Welfare Toolkit where it can be adapted as per the collection who wishes to use it, therefore we are trying to not be prescriptive with how it can be implemented. We have included information for private keepers that the statistical analysis is not essential, it is the review of what they see and how they collect information in a standardised and methodical manner that is important.

Appendix 1: Item 3. Typo, “in place THAT provides…”.

Thank you. We have corrected this.

Lines 254-259, Lines 270-277. In section 2.1, the authors discuss that natural behaviour offers a good starting point to consider welfare, but that in captivity, some birds may not need to display natural behaviour in a captive environment where they don’t encounter the same challenges as in the wild. So I was a bit disappointed that lines 254-259 and 270-277 give the impression that improving welfare is simply about letting the birds do their natural behaviour. I don’t think this was your intention, but it comes across like that when I read it. Perhaps you could re-consider writing these sections to make clear that simply allowing animals to do natural behaviour may not be necessary to ensure good welfare.

Thank you for the comments. We have edited this to improve clarity, based on your recommendations to explain that a lack of performance of some natural behaviours does not always equate to poorer welfare. We have gone back through the paper to clarify where we explain natural behaviour to ensure that it is in context to what the animal is responding to.

Reviewer 2 Report

This ambitous paper addresses an important area of animal welfare in an overlooked taxon. The authors have collated a substantive amount of published and empirical information on anseriformes welfare and advance a suggested evaluation system through a protocol. They also address salient aspects of anseriform welfare such as pinioning.

However, this article falls short of providing a synthetic, consistent explanation of the rationale for developing the proposed protocol and its objective validation, as wella as guidance on how the results are to be evaluated.

Shortcomings include:

  • the text needs to be streamlined, and have a clearer structure. In many instances, the authors have dropped to a narrative presentation rather than a scientific description of concepts and discussion.
  • authors need to revise key animal welfare concepts, such as what resource-based vs animal-based indicators are (e.g. lines 29-31)
  • the manuscript needs to provide a clear theoretical underpinning for the proposed protocol. Reference to alternative welfare assessment theoretical frameworks is wanting. Review of other bird welfare assessment protocols needs to be included.
  • the authors fail to provide or make a proposal for the validation of the protocol, i.e. how to ensure that the protocol results actually relate to animal welfare
  • the manuscript does not provide clear usage guidelines for the protocol: how much observation time per animal is required, how it is to be applied in flocks vs individual animals

As regards the protocol itself, it needs to be thoughroughly revised. Although all the questions can arguably relate to animal welfare (e.g. moult state or presence of public) there is no indication on how these are to be interpreted as regards welfare. Similarly, the proposed behavioural observations, albeit interesting, require a guidance to be interpreted if they are to be used as welfare indicators.

A re-written manuscript advancing the proposed protocol needs to include:

  1. a clear reference to existing theoretical frameworks for welfare assessment
  2. reference to existing validated welfare assessment proposals in birds
  3. the rationale for each of the selected indicators & scope for application of the protocol
  4. discussion on how the proposed protocol (as a result) relates or compares to other welfare protocols
  5. clarity on the validation system advanced in figure 3: this appears to be a worthwhile validation of subjective behavioral observation
  6. discussion on how objective validation of the protocol is to be achieved
  7. clear guidelines for how to implement the protocol in the anticipated settings

[there is also need for English proofreading for gramatical errors and typos]

Author Response

Reviewer 2

This ambitous paper addresses an important area of animal welfare in an overlooked taxon. The authors have collated a substantive amount of published and empirical information on anseriformes welfare and advance a suggested evaluation system through a protocol. They also address salient aspects of anseriform welfare such as pinioning. However, this article falls short of providing a synthetic, consistent explanation of the rationale for developing the proposed protocol and its objective validation, as wella as guidance on how the results are to be evaluated.

Thank you for the comments. We have addressed many of the points that you raise here in other reviewer comments, but we have revisited the manuscript and have added in some key details that enable the paper to show how this method of welfare scoring could work and why it is applicable to these species of bird. We have included more information on how to conduct a welfare assessment using the tool provided, but we are not trying to be prescriptive. With the information that we were provided with when developing this method, it was made clear to provide an example of a method that could be tailored to the facilities who wish to use it. And that what we have produced. We hope that the edits to the section on using the tool, developing a way of using the information from tool, help clarify areas of uncertainty.

Shortcomings include:

the text needs to be streamlined, and have a clearer structure. In many instances, the authors have dropped to a narrative presentation rather than a scientific description of concepts and discussion.

Thank you for the comments. We have attempted to include remove sections that are not useful (based on other reviewer comments) and we have included more information on how to conduct a welfare assessment. We have included more scientific evaluation of what welfare is, and what welfare assessment is, and we have expanded on the use of the tool towards the end of the paper.

We have a narrative of key animal behaviour patterns as they are important to describe when explaining why need to be measure welfare. It is important for keepers and animal managers to see the link between what the animal is doing / would like to do / does not need to do and what it is provided with in captivity.

authors need to revise key animal welfare concepts, such as what resource-based vs animal-based indicators are (e.g. lines 29-31)

Thank you for the comment. We do provide an explanation of what these mean in the abstract but we have expanded on what this means in our introduction to show how this links to what is being measured in an animal welfare assessment. We have included some new sources that further explain and expand upon what these types of approaches are (resource based and animal based).

the manuscript needs to provide a clear theoretical underpinning for the proposed protocol. Reference to alternative welfare assessment theoretical frameworks is wanting. Review of other bird welfare assessment protocols needs to be included.

Thank you for the suggestions. We have re-structured the introduction to show the link between the explanation of what welfare is and the background to animal welfare assessment and the reason why species-specific measurement approaches (i.e. our attempt at an Anseriformes model) are needed. Hopefully this provides more clarity and evidence for the assessment tool.

We have explained welfare assessment that is used in domestic poultry but why this is not a good baseline for zoo birds. And we evaluate the one paper that we can find that does include some attempts at scoring zoo bird welfare. We set the scene with the BIAZA Animal Welfare Toolkit that forms the basis for why this paper was written (and is needed).

the authors fail to provide or make a proposal for the validation of the protocol, i.e. how to ensure that the protocol results actually relate to animal welfare

Thank you for the comment. We have included more information on how to implement the welfare assessment based on review and re-evaluation of what the methods can show the reader. We feel that we have explained this as we have a section that not only related to use of the information gained, but we show an example of how to use it (Figure 5) and a detailed reasoning for why this cycle of re-evaluation helps provide for good welfare (Figure 6).  

the manuscript does not provide clear usage guidelines for the protocol: how much observation time per animal is required, how it is to be applied in flocks vs individual animals

Thank you for the comment. We have provided some more information on this aspect of using the welfare assessment methods.

As regards the protocol itself, it needs to be thoughroughly revised. Although all the questions can arguably relate to animal welfare (e.g. moult state or presence of public) there is no indication on how these are to be interpreted as regards welfare. Similarly, the proposed behavioural observations, albeit interesting, require a guidance to be interpreted if they are to be used as welfare indicators.

We feel that these descriptors and explanation of provision (husbandry and enclosure features) and animal based outputs (behavioural characteristics) are reflective of animal welfare. They were reviewed by industry experts prior to submission and we have amended them based on their feedback and on how we have used them in practice. We have provided more information on this review process and how the information taken from it can be used. We have described each stage of the methods (section 4.3) and what they tell a keeper and why this factor is being evaluated. We have also written two new sections to provide further assistance on implanting welfare scoring. We have provided guidance on the interpretation of the welfare indicators on a species-specific level.

A re-written manuscript advancing the proposed protocol needs to include:

a clear reference to existing theoretical frameworks for welfare assessment

Thank you for the suggestion. We have included more background information.

reference to existing validated welfare assessment proposals in birds

Thank you for the suggestion. We have included papers that we can find but if you are aware of any other zoo bird specific welfare assessment tools, please let us know and we can include also.

the rationale for each of the selected indicators & scope for application of the protocol

We have expanded on the pre-existing explanation of this.

discussion on how the proposed protocol (as a result) relates or compares to other welfare protocols

We have provided more explanation on the need for this welfare assessment tool and how it links back to industry.

clarity on the validation system advanced in figure 3: this appears to be a worthwhile validation of subjective behavioral observation

We have included more information on these statistical outputs with the proviso that we would not expect a private animal keeper to do this. This could be undertaken by a zoo’s research department or animal welfare scientist if they wished to see any correlations between welfare descriptors and husbandry over time.

discussion on how objective validation of the protocol is to be achieved

We have explained how to perform interobserver reliability and measure the validity of the welfare scoring methods provided. At the request of another reviewer we have also explained key areas of the toolkit that should be considered based on the specific species being observed. And we have explained how to link together different sections of the toolkit should there be different outcomes from each.

clear guidelines for how to implement the protocol in the anticipated settings

We have included more information on this to show how it is relevant across taxa.

[there is also need for English proofreading for gramatical errors and typos]

Thank you. We have read over the manuscript and attempted to correct all typographical errors.

Reviewer 3 Report

This manuscript provides a method for measuring the welfare of waterfowl for zee-housed birds and the private kept birds.  Authors introduced the background of the needs of waterfowl welfare and the purpose of this manuscript in very deep detail.   They also described the tool and the guidance for using it for the assessment.  On the whole, the structure, objective and description of this manuscript are good enough to be published as it.

Author Response

Reviewer 3

This manuscript provides a method for measuring the welfare of waterfowl for zee-housed birds and the private kept birds.  Authors introduced the background of the needs of waterfowl welfare and the purpose of this manuscript in very deep detail.   They also described the tool and the guidance for using it for the assessment.  On the whole, the structure, objective and description of this manuscript are good enough to be published as it.

Thank you for the comments. We appreciate that you have enjoyed the paper and found it useful and interesting.

Round 2

Reviewer 1 Report

The authors have addressed all my comments and I found this revised paper to be more robust, and have no further major comments.

Line 233. Both authors? In which case it should be “authors”.

Line 294. Check sentence, “simply” seems out of place.

Line 424. Remove “is”.